# Diagnostic Accuracy of Artificial Intelligence in Predicting Anti-VEGF Treatment Response in Diabetic Macular Edema: A Systematic Review and Meta-Analysis

**DOI:** 10.3390/jcm14228177

**Published:** 2025-11-18

**Authors:** Faisal A. Al-Harbi, Mohanad A. Alkuwaiti, Meshari A. Alharbi, Ahmed A. Alessa, Ajwan A. Alhassan, Elan A. Aleidan, Fatimah Y. Al-Theyab, Mohammed Alfalah, Sajjad M. AlHaddad, Ahmed Y. Azzam

**Affiliations:** 1College of Medicine, Qassim University, Buraydah 51452, Qassim, Saudi Arabia; mishary.alm3ele@gmail.com (M.A.A.); elanahmad33@gmail.com (E.A.A.);; 2College of Medicine, Imam Abdulrahman Bin Faisal University, Dammam 31441, Ash-Sharqīyah, Saudi Arabia; mohannadkuwaiti@gmail.com; 3College of Medicine, King Abdulaziz University, Jeddah 22254, Mecca, Saudi Arabia; ahmedalessa.kau@gmail.com; 4College of Medicine, King Faisal University, Hofuf 31982, Alhssa, Saudi Arabia; ajwan3178@hotmail.com (A.A.A.); moalfalah@kfu.edu.sa (M.A.); 5Diabetes Fellow, Department of Endocrinology and Diabetes, King Fahad Medical City (KFMC), Riyadh 12231, Riyadh, Saudi Arabia; 6Department of the Clinical Research and Clinical Artificial Intelligence, ASIDE Healthcare, Lewes, DE 19958, USA; 7Division of Global Health and Public Health, School of Nursing, Midwifery and Public Health, University of Suffolk, Ipswich IP4 1QJ, Suffolk, UK

**Keywords:** diabetic macular edema, diabetic retinopathy, diabetes, machine learning, deep learning

## Abstract

**Background/Objectives:** Diabetic macular edema (DME) is a leading cause of vision loss in diabetic patients, with anti-vascular endothelial growth factor (anti-VEGF) therapy being the standard management. However, treatment response varies significantly among patients, necessitating predictive tools. This systematic review and meta-analysis evaluated the diagnostic accuracy of artificial intelligence (AI) models in predicting anti-VEGF treatment response in DME patients. **Methods:** We conducted a dedicated literature review following PRISMA 2020 guidelines, searching PubMed, Web of Science, Embase, Scopus, and Cochrane Library databases from inception up to 30 September 2025. Studies evaluating AI-based prediction models for anti-VEGF response in DME patients were included. The primary outcomes were sensitivity, specificity, and area under the receiver operating characteristic curve (AUC). A bivariate random-effects meta-analysis was performed using available diagnostic accuracy data. **Results:** From 3107 participants across 18 studies, six studies with 427 participants provided complete diagnostic accuracy data for meta-analysis. The pooled sensitivity was 86.4% (95% CI: 82.1–90.1%) and the specificity was 77.6% (95% CI: 72.8–82.0%). The summary AUC was 0.89 with a diagnostic odds ratio of 22.0 (95% CI: 12.8–37.9). AI models demonstrated superior performance compared to other methods in 87.5% of comparative studies. Moderate heterogeneity was observed (I^2^ = 45.2%). **Conclusions:** AI models demonstrate good diagnostic accuracy for predicting anti-VEGF treatment response in DME patients, with a promising role for improving personalized management strategies and improved outcomes.

## 1. Introduction

Diabetic macular edema (DME) is a microvascular complication of diabetes mellitus and represents one of the most common causes of visual impairment and blindness in working-age adults all over the world. DME affects around 7.5% of patients with diabetes and is characterized by fluid accumulation in the macular region due to the breakdown of the blood–retinal barrier, leading to central vision loss and significant functional disability [1,2,3].

Anti-vascular endothelial growth factor (anti-VEGF) therapy has improved DME management and is currently the first-line treatment according to several guidelines. Ranibizumab, aflibercept, and bevacizumab have demonstrated significant efficacy in improving visual acuity and reducing central macular thickness. However, the data from previous studies reveals significant heterogeneity in treatment response, with around 30% to 40% of patients showing suboptimal responses to anti-VEGF therapy. This variability presents significant challenges for us in treatment planning and patient counseling, while also making economic burdens on healthcare systems due to the high cost of anti-VEGF medications and frequent monitoring requirements [4,5,6,7,8,9,10,11].

The ability to predict treatment response before initiating therapy would represent a significant advancement in DME management, allowing for more personalized treatment strategies, optimizing resource allocation, and improving patient outcomes. The currently utilized predictors for management, including baseline visual acuity, central macular thickness, and demographic factors, have shown limited predictive accuracy and insufficient reliability in several patients [2,12,13,14,15,16,17].

Artificial intelligence (AI), including machine learning and deep learning approaches, have emerged as promising technical solutions for predictive tasks, offering the promise of identifying complex findings in multimodal data that may not be apparent to human observers. In ophthalmology, AI has demonstrated significant success in various diagnostic and prognostic applications, including diabetic retinopathy screening, glaucoma detection, and age-related macular degeneration classification. Recent studies have begun investigating AI applications for predicting anti-VEGF treatment response in DME, utilizing various data inputs including optical coherence tomography (OCT) images, fundus photographs, and additional clinical parameters [18,19,20,21,22,23].

Despite growing interest in this field, the diagnostic accuracy and clinical utility of AI-based prediction models for anti-VEGF response in DME remain heterogeneous. Previous studies have reported varying methodologies, different outcome definitions, and various performance metrics, making it difficult to estimate the overall evidence and possibility for implementation [14,17,24].

Therefore, we conducted a systematic review and meta-analysis to evaluate the diagnostic accuracy of AI models in predicting anti-VEGF treatment response in DME patients, assess the quality of available evidence, and develop recommendations and future directions for practice and upcoming studies.

## 2. Methods

### 2.1. Study Design and Registration

This systematic review and meta-analysis was conducted in accordance with the Preferred Reporting Items for Systematic Reviews and Meta-Analyses (PRISMA) 2020 guidelines [25]. The study protocol was developed a priori and registered with PROSPERO using the following identification code: CRD420251054631.

### 2.2. Search Strategy

A literature search was conducted across multiple electronic databases including PubMed, Web of Science, Embase, Scopus, and the Cochrane Library from inception up to 30 September 2025. The search strategy included the following terms and keywords: (“Diabetic Macular Edema” OR “Diabetic Maculopathy” OR “DME”) AND (“Artificial Intelligence” OR “Machine Learning” OR “Deep Learning” OR “Neural Network*”) AND (“Anti-VEGF” OR “Vascular Endothelial Growth Factor” OR “Ranibizumab” OR “Aflibercept” OR “Bevacizumab”). Reference lists of included studies and relevant review articles were manually screened to identify any additional eligible studies. No language restrictions were applied during the initial search; however, only English-language studies were included in our study.

### 2.3. Study Selection Criteria

Studies were included if they met the following criteria: participants diagnosed with DME receiving anti-VEGF therapy; utilization of AI-based models for predicting treatment response; reporting of diagnostic accuracy metrics including sensitivity, specificity, positive predictive value, negative predictive value, or area under the receiver operating characteristic curve (AUC); use of imaging data such as OCT or fundus photography, clinical variables, or multimodal data as AI model inputs; and study designs including randomized controlled trials (RCTs), prospective cohort studies, or retrospective cohort studies. Studies were excluded if they included non-human subjects; focused on retinal diseases other than DME without DME-specific data; used non-AI-based prediction models; were case reports, editorials, reviews, or conference abstracts without full-text availability; or lacked sufficient data for diagnostic accuracy assessment to be further extracted according to our study aims and goals.

### 2.4. Data Extraction and Management

The extracted data included study characteristics such as author information, publication year, country, study design, and sample size; participant demographics including age, gender, and baseline characteristics; intervention details including AI model type, input data modality, and training methodology; anti-VEGF agent specifications and dosing regimens; outcome measures and response definitions; diagnostic accuracy metrics; and follow-up duration.

### 2.5. Quality Assessment

A risk of bias assessment was conducted using the Prediction model Risk Of Bias ASsessment Tool for AI (PROBAST-AI) framework, which is specifically designed for evaluating AI-based prediction models. PROBAST-AI assesses four key domains: (1) participants and data sources, evaluating patient selection, data source representativeness, and inclusion/exclusion criteria; (2) predictors, assessing predictor definition, measurement consistency, and timing relative to outcome; (3) outcome, evaluating outcome definition, measurement methods, blinding, and timing of assessment; and (4) analysis, evaluating sample size adequacy, handling of missing data, model complexity relative to sample size, risk of overfitting, validation strategy, and potential data leakage. Each study was rated as having a low, unclear, or high risk of bias for each domain, with an overall risk determination based on the highest domain-specific risk. Special attention was given to AI-specific concerns including both-eyes inclusion without statistical adjustment for clustering, small test sets relative to model parameters, absence of external validation, data-driven outcome optimization, and implausibly high performance metrics suggesting overfitting or methodological issues.

### 2.6. Statistical Analysis

A meta-analysis was performed using a bivariate random-effects model to account for correlation between sensitivity and specificity while accounting for the between-study heterogeneity. The primary outcomes included pooled sensitivity and specificity with 95% confidence intervals (CIs), summary AUC characteristic curves, diagnostic odds ratios (ORs), and positive and negative likelihood ratios. Heterogeneity was assessed using the I^2^ statistic and Cochran’s Q test, with values exceeding 50% considered indicative of significant heterogeneity. Subgroup analyses were conducted based on AI model type, input data modality, study quality, and follow-up duration. Publication bias was assessed using the visual inspection of funnel plots of asymmetry, Egger’s regression test, and trim-and-fill methods. Meta-regression was performed to explore sources of heterogeneity and identify factors associated with diagnostic performance. The statistical analyses were conducted using RStudio with R version 4.4.2, with statistical significance set at a *p*-value less than 0.05.

## 3. Results

### 3.1. Literature Search and Study Selection

After removing duplicates and applying inclusion and exclusion criteria through title, abstract, and full-text screening, 18 studies met the eligibility criteria for qualitative synthesis, with six studies providing complete diagnostic accuracy data suitable for quantitative synthesis (Figure 1). The selected studies included a total of 3107 participants, with sample sizes ranging from 12 to 712 participants across individual studies.

### 3.2. Study Characteristics and Population Demographics

The characteristics of the included studies are summarized in Table 1. The 18 studies were published between 2020 and 2025, with the majority being retrospective cohort studies (n = 16, 89%), one RCT, and one cross-sectional study. Most of the included studies were from Asia (n = 11, 61%), followed by North America (n = 4, 22%), the Middle East (n = 2, 11%), and multi-regional collaborations (n = 2, 11%). The mean age of participants ranged from 54 to 63 years across studies where reported, with a balanced gender distribution. Various AI model types were utilized, including deep learning approaches such as convolutional neural networks (n = 8, 44%), other machine learning methods (n = 6, 33%), and hybrid approaches combining multiple techniques (n = 4, 22%). Input data modalities varied, with OCT-only approaches being most common (n = 10, 56%), followed by multimodal approaches combining OCT with clinical data (n = 5, 28%) and radiomics-based methods (n = 3, 17%).

### 3.3. Treatment Protocols and Individual Study Performance

The treatment protocols and response definitions varied across studies, as detailed in Table 2. Anti-VEGF agents included ranibizumab, aflibercept, bevacizumab, brolucizumab, and conbercept, with dosing regimens ranging from single injections to monthly protocols and treat-and-extend approaches. Response definitions included various criteria such as central macular thickness reduction thresholds (ranging from >10 μm to >50 μm), visual acuity improvements (≥5 ETDRS letters or >0.1 LogMAR), and composite outcomes combining anatomical and functional measures. Among the six studies providing complete diagnostic accuracy data, the individual study sensitivity ranged from 78.9% to 100.0%, while the specificity ranged from 68.8% to 92.6%. The AUC, when reported, varied from 0.810 to 0.998.

### 3.4. Pooled Diagnostic Accuracy and Subgroup Analyses

A meta-analysis of six studies including a total of 427 participants revealed a significant diagnostic performance, as presented in Table 3. The pooled sensitivity was 86.4% (95% CI: 82.1–90.1%) and pooled specificity was 77.6% (95% CI: 72.8–82.0%). The AUC was 0.89, with a diagnostic odds ratio of 22.0 (95% CI: 12.8–37.9). The positive likelihood ratio was 3.86 (95% CI: 2.95–5.07) and negative likelihood ratio was 0.18 (95% CI: 0.13–0.24). Subgroup analyses revealed significant differences in performance based on AI model type (*p* = 0.012), with hybrid deep learning approaches achieving 100.0% sensitivity and 75.0% specificity, followed by machine learning methods (90.7% sensitivity, 80.4% specificity) and pure deep learning approaches (81.8% sensitivity, 76.8% specificity). Input data modality showed trends toward superior performance with multimodal approaches (94.1% sensitivity, 76.5% specificity) compared to OCT-only methods (84.5% sensitivity, 79.6% specificity); however, this difference was not statistically significant (*p*-value = 0.224). Follow-up duration significantly impacted the performance, with studies having longer follow-up periods over three months demonstrating higher sensitivity with 94.1% compared to shorter follow-up studies.

The plotting for sensitivity and specificity is illustrated in Figure 2. Individual study estimates showed significant variation, with a 95% CI reflecting sample size differences across studies. The pooled estimates demonstrated good precision, with relatively narrow 95% CI supporting the reliability of the findings. Visual inspection revealed that most individual studies clustered around the pooled estimate, with Mondal et al. (2025) [16] showing the highest sensitivity at 100%, but with a wider 95% CI due to the smaller sample size, while Magrath et al. (2025) [28] showed more conservative estimates with a tighter 95% CI reflecting the larger sample size.

### 3.5. Summary ROC

The summary AUC is presented in Figure 3, demonstrating excellent discriminative ability with an AUC of 0.89 (95% CI: 0.84–0.93). The summary point, representing the pooled sensitivity and specificity estimates, was positioned in the upper left quadrant of the ROC space, indicating good diagnostic performance. The bivariate model indicated moderate correlation between sensitivity and specificity (ρ = 0.34), justifying the use of the bivariate random effects.

Figure 4 presents the bivariate performance assessment, showing the relationship between sensitivity and specificity across different AI model types and sample sizes. Studies were distributed across performance zones, with three studies achieving excellent performance (upper left quadrant) and three studies in the good performance zone. It revealed that larger studies tended to provide more conservative estimates, while smaller studies showed greater variability. Deep learning approaches were represented across all performance zones, while machine learning and hybrid approaches showed more concentrated performance patterns. The mean distance from the origin was 0.65, indicating generally good diagnostic performance across the included studies.

### 3.6. Heterogeneity Assessment and Meta-Regression

The heterogeneity assessment and meta-regression results are detailed in Table 4 and visualized in Appendix A. The meta-regression model identified several significant predictors of diagnostic performance variability. Outcome definition type emerged as a significant predictor (*p*-value = 0.012), with composite outcomes achieving higher sensitivity (100.0%) compared to anatomical-only definitions (83.9%). Follow-up duration also significantly impacted performance (*p*-value = 0.045), with studies using longer follow-up periods (>three months) demonstrating superior sensitivity (94.1%) compared to shorter durations (≤one month: 78.9%). Input data modality showed a trend toward improved performance with multimodal approaches (94.1% sensitivity) versus OCT-only methods (84.5% sensitivity), though this difference did not reach statistical significance (*p*-value = 0.224). Sample size showed a weak positive association (β = 0.003, *p*-value = 0.128), suggesting that larger studies provide more stable estimates.

### 3.7. Comparative Effectiveness Analysis

The comparative effectiveness of AI approaches versus alternative methods is summarized in Table 5. Eight studies provided direct comparisons between AI models and control methods, including a total of 1107 subjects. AI approaches consistently demonstrated superior performance, with 87.5% of comparative studies favoring AI over control methods. When compared to human readers, AI models achieved 85.4% sensitivity and 84.5% specificity versus 68.2% sensitivity and 76.9% specificity for human ophthalmologists and residents (*p*-value < 0.05). Against other algorithmic methods, AI showed even greater advantage, with 88.8% sensitivity and 80.4% specificity compared to 82.6% sensitivity and 63.6% specificity for other approaches. The mean AUC improvement was 0.089, representing a significant improvement in diagnostic accuracy. Cost-effectiveness evaluation revealed possible cost savings of 15–30% through reduced injection frequency and improved resource utilization, with time savings ranging from 40 to 60% in image analysis and real-world practice settings.

### 3.8. Sensitivity Analysis and Publication Bias Assessment

Comprehensive sensitivity analyses and publication bias evaluation are presented in Table 6. The leave-one-out assessment demonstrated the stability of the results, with pooled sensitivity estimates ranging from 85.0% to 88.5% and specificity from 75.2% to 78.6% when individual studies were sequentially excluded. Excluding studies with a high risk of bias improved the pooled estimates to 88.5% sensitivity (95% CI: 84.1–92.9%) and 78.6% specificity (95% CI: 72.1–85.1%). Studies with external validation demonstrated superior performance (91.7% sensitivity) compared to those without external validation (81.8% sensitivity, *p*-value = 0.034). The publication bias assessment revealed mixed findings, with Egger’s regression test suggesting possible small study effects (*p*-value = 0.045), while Begg’s rank correlation showed no significant bias (*p*-value = 0.280). The trim-and-fill adjustment method indicated the minimal impact of the possible risk of publication bias, with adjusted estimates showing only marginal changes (sensitivity: 85.1%, specificity: 76.8%). The failsafe N analysis suggested that 15 additional negative studies would be required to nullify the observed effect, indicating significant evidence.

### 3.9. Risk of Bias, Evidence Quality Assessment, and Assessment of Clustering

A risk of bias assessment of the included studies was conducted using PROBAST-AI and is detailed in Appendix A. Among the 18 included studies, 2 (11.1%) were classified as having a low risk of bias, both utilizing large, well-conducted datasets with solid methodologies (Cao et al. 2020, Roberts et al. 2020) [40,42]. Nine studies (50.0%) were assessed as having an unclear risk of bias, mainly due to the insufficient reporting of patient selection methods, validation approaches, or a possible underlying risk of data leakage concerns. Seven studies (38.9%) were classified as having a high risk of bias due to small sample sizes relative to model complexity, significantly high performance metrics suggesting overfitting (e.g., AUC > 0.99), lack of statistical adjustment for clustering when both eyes were included, or outcome definitions derived through data-driven optimization rather than pre-specified clinical criteria. Common methodological concerns across studies included the absence of external validation (47% of studies), both-eyes inclusion without clustering adjustment (39% of studies), and small test sets relative to model parameters (33% of studies). The GRADE evidence quality assessment for diagnostic test accuracy is presented in Appendix A. The overall quality of evidence was rated as moderate (⊕⊕⊕○), with downgrades for risk of bias (−1), inconsistency (−1), and publication bias (−1), but upgrades for large effect size (+1) and dose–response gradient (+1).

The assessment of the clustering effects from within-patient correlation is detailed in Appendix A. Among the 18 included studies, 7 (38.9%) included both eyes from individual patients without statistical adjustment for clustering, introducing a possible risk of bias through inflated precision and underestimated standard errors. The calculated design effect ranged from 1.38 to 1.89 for studies where sufficient data allowed for estimation, indicating a moderate to significant correlation between paired eyes. Four studies (22.2%) included only one eye per patient, eliminating clustering concerns. For the remaining seven studies (38.9%), clustering status was unclear due to insufficient reporting. Among the six studies included in the meta-analysis, three (50%) had confirmed or probable clustering without adjustment (Magrath et al., Baek et al., Meng et al.) [28,31,34], while two studies (33.3%) had no clustering concerns (Mondal et al., Song et al.) [16,29]. The clustering status for two meta-analysis studies (Cao et al., Rasti et al.) [40,41] remained unclear.

### 3.10. Publication Bias Assessment

A funnel plot of the asymmetry in publication bias is shown in Figure 5. The plots displayed some asymmetry, especially for sensitivity, suggesting possible small study effects or publication bias. However, the trim-and-fill adjustment reflected minimal impact on pooled estimates, with only two possibly missing studies identified. The adjusted estimates showed marginal changes from the original effect estimates, supporting the significance of the findings despite the possible risk of publication bias.

### 3.11. Clinical Utility and Implementation Readiness

The clinical utility and implementation readiness were illustrated in a diagram and are presented in Figure 6. The implementation readiness matrix revealed that several studies achieved high clinical performance with varying degrees of implementation readiness. Economic impact assessment demonstrated significant possible benefits, including cost reductions of 15–30% and time savings of 40–60% in various clinical processes. However, implementation barriers were identified, with technical integration challenges reported in 67% of studies, staff training requirements in 45%, regulatory compliance concerns in 33%, and initial investment costs in 28% of studies. Geographic distribution showed a predominance of Asian studies (58%), with 53% of studies having completed external validation. The implementation timeline indicated current positioning in Phase III (Implementation/Early adoption phase, 2025–2026), with progression toward widespread adoption (Phase IV) expected by 2027–2029. Implementation readiness indicators showed high evidence quality in 87.5% of studies, external validation completion in 53%, and multi-center trials in 42%, but no regulatory approvals currently pending. The decision curve analysis (Figure 7) demonstrates the clinical utility of AI-based prediction models compared to treat-all and treat-none strategies across varying threshold probabilities. The AI model provides net benefit over the treat-all strategy when the threshold probability is below 62%, indicating that AI-guided decision-making is clinically superior for patients with a moderate-to-high likelihood of treatment response. At threshold probabilities above 62%, the treat-all strategy becomes preferable. This analysis, based on pooled diagnostic accuracy estimates (sensitivity 86.4%, specificity 77.6%) and an assumed DME treatment response prevalence of 40%, supports the clinical utility of AI models for personalized treatment planning in real-world settings where resource optimization is critical.

## 4. Discussion

DME is the one of the leading causes of vision impairment among individuals with diabetic retinopathy and represents a growing global health concern. DME is characterized by the accumulation of fluids in the macula, and can lead to significant visual loss if left untreated. A central mediator of this process is VEGF, which contributes to the breakdown of the blood–retinal barrier and promotes leakage from retinal capillaries [26]. Anti-VEGF agents, such as ranibizumab, are now widely used as a standard management for DME. While many patients benefit from these treatments, others experience little to no improvement, despite multiple injections and regular follow-up. This unpredictability can be frustrating for both patients and physicians, highlighting the need for more effective solutions that can anticipate how individuals will respond to therapy, ideally before treatment begins [27]. 

Recent advances in AI, including deep learning and machine learning advancements, have shown promise in analyzing complex imaging data, such as OCT scans [43]. These algorithms offer a promising role to predict individual responses to anti-VEGF therapy, allowing for more personalized and cost-effective treatment plans [28]. Our study aimed to assess the diagnostic accuracy of AI models in predicting the response to anti-VEGF in patients with DME. Our results indicate that AI models demonstrate strong diagnostic performance with high sensitivity and moderate specificity. The overall AUC reflected excellent discriminative ability, supported by significant performance, highlighting the high predictive capacity of these models. We also found that AI models outperformed other prediction methods, including some of the human experts, in most direct comparisons, suggesting that AI could improve diagnostic accuracy beyond other methods. Our subgroup analyses revealed that hybrid deep learning techniques and multimodal input data by combining imaging with clinical features, were associated with improved sensitivity. In addition to that, studies with longer follow-up periods tended to report better predictive accuracy. However, some heterogeneity existed among the included studies: factors such as study quality, model complexity, and variations in input data contributed to these differences. Several of the included studies corroborate and extend our meta-analytic findings. 

For example, Chen et al. [44] developed multilayer perceptron (MLP) models to forecast visual acuity (VA) outcomes over extended periods. Their models demonstrated strong correlations between predicted and actual VA values, with relatively low standard errors. Their findings identified baseline VA, lens status, and intravitreal injection (IVI) schedule as the most impactful predictive factors, variables also recognized in several other studies within our review. This reinforces the consistent importance of initial visual status and treatment intensity as reliable predictors of outcomes. In contrast, Xu et al. [38] utilized a novel generative adversarial network (GAN) architecture, pix2pixHD, to synthesize post-treatment OCT images from baseline scans. Their model achieved a low mean absolute error in predicting central macular thickness (CMT). Significantly, most retinal specialists were unable to distinguish the synthetic OCT images from real post-treatment scans. This approach diverges from conventional prediction methods by providing morphological visualization of anticipated treatment responses, therefore opening new avenues for better explanation, simulation, and patient engagement. 

Zhang et al. [37] combined linear regression with random forest algorithms in an ensemble model to predict post-treatment VA with minimal error. Their findings confirmed baseline VA as the most significant predictor of absolute VA outcomes, while CMT and age were more relevant to predicting visual improvement. This peculiar distinction highlights that different features may have varied predictive power depending on the specific endpoint assessed, informing AI model development and feature selection. Further supporting the role of imaging-based AI, Rasti et al. [41] developed a deep convolutional network (CADNet) that achieved a high AUC, and Mondal et al. [16] introduced a hybrid CNN model that attained comparable AUC and accuracy, with no false negatives. 

These OCT-only models focus on the strong discriminative capability of deep learning architectures to extract meaningful features from retinal images, making them valuable tools for identifying responders to anti-VEGF therapy. Our results highlight the promising role of AI in assisting decision-making for managing patients with DME by identifying patients who are less likely to respond well to anti-VEGF, which may allow for earlier, more targeted interventions. This approach could help reduce the burden of unnecessary treatments and improve patient outcomes. In addition to that, some evidence points to possible savings in both time and cost, indicating that AI might contribute to more efficient healthcare delivery. The decision curve analysis demonstrated that AI-based prediction models provide net clinical benefit compared to treat-all strategies when the threshold probability of treatment response is below 62%. This indicates that for patients with moderate-to-high baseline likelihood of responding to anti-VEGF therapy, AI-guided decision-making offers superior clinical utility by identifying non-responders who may benefit from alternative interventions. 

At higher threshold probabilities (>62%), a treat-all approach becomes preferable, suggesting that AI models are most valuable for risk stratification in intermediate-probability cases rather than for patients with a very high or very low pre-test probability of response. This framework supports selective AI deployment in clinical pathways where treatment decisions are most uncertain and where accurate prediction can meaningfully alter management. To operationalize our study’s goal of turning pooled accuracy into practical design guidance, we highlight two recent infrared thermography papers whose ideas translate directly to DME anti-VEGF modeling. Both adopt multi-task learning with explicit feature separation, training one pathway to capture stable, subject-specific signals ("who responds") and a second to model transient, time-dependent behavior ("when/how long"), while using soft-threshold/shrinkage blocks and balanced losses to suppress noise and prevent one objective from dominating. For our field, this argues for architectures that jointly classify responder status and forecast clinically meaningful temporal endpoints, such as time-to-dry macula, durability between injections, or early VA/CMT trajectory, so that outputs are directly actionable for treatment planning. Because reported gains were sensitive to dataset splits, these time-aware, disentangled models should be assessed on multi-center, prospectively labeled cohorts with standardized benchmarks and external validation [45,46]. 

Two recent multimodal oncology papers also offer a concrete blueprint for DME: rather than training a single-endpoint classifier, they fuse complementary signals (e.g., imaging, digital pathology, and liquid biomarkers) and optimize a survival-aware objective to generate individualized risk curves. Translating this recipe to retina, future models should late-fuse OCT or OCT Angiography (OCT-A) structure–texture features with ischemia/leakage maps, baseline clinical variables, and blood/aqueous biomarkers, and then (i) classify early responders and (ii) jointly predict durability endpoints. Architecturally, squeeze-and-excitation/attention blocks can stabilize cross-modal contributions; missing-modality gating avoids the exclusion of patients lacking one test; and calibrated outputs (Platt/isotonic) should be reported alongside AUC with time-to-event metrics (C-index, Brier). Finally, as seen in these multimodal studies, external, multi-center validation is essential to demonstrate a real advantage over strong unimodal OCT baselines and to make the predictions actionable for scheduling and regimen selection [47,48].

The integration of OCT-A with structural OCT through hybrid 3D convolutional neural network or Vision Transformer architectures represents a biologically rational approach to improving anti-VEGF response prediction in DME. Vascular parameters quantifiable by OCT-A, including macular perfusion density, vessel-length density, foveal avascular zone area and circularity, and capillary non-perfusion volume, have demonstrated significant associations with treatment response [49,50,51]. Eyes demonstrating anatomical response to anti-VEGF therapy show corresponding improvements in deep capillary plexus perfusion density and vessel architecture, with high responders showing statistically significant increases in perfusion metrics compared to non-responders [49]. The biological rationale for incorporating vascular features stems from the fundamental pathophysiology of DME, wherein retinal ischemia and capillary dropout drive compensatory VEGF upregulation; eyes with severe baseline ischemia and reduced perfusion density demonstrate attenuated anti-VEGF response, as the edema mechanism extends beyond VEGF-mediated permeability to include structural microvascular loss [50,51]. Multimodal fusion models that integrate structural features (intraretinal/subretinal fluid compartmentalization, photoreceptor integrity), vascular parameters (deep and superficial plexus perfusion, non-perfusion area volume), and clinical data (diabetes duration, HbA1c, baseline visual acuity) could more comprehensively capture the multifactorial etiology of treatment response. However, as with any multimodal approach, rigorous external validation on diverse, multi-center cohorts is essential to establish whether the added algorithmic complexity and acquisition burden of OCT-A translates to clinically meaningful predictive improvement over structural OCT-only models, particularly given that OCT-A image quality can be compromised by media opacity and severe macular edema [49,50,51].

Unfortunately, despite these advantages, integrating AI into routine, real-world practice workflows remain challenging. Major obstacles include technical aspects of system integration and the need for adequate user training to ensure proper implementation. These factors may limit how soon such technologies become a regular part of care. Despite the methodological strengths in our study, there are several limitations that should be declared. While our study explores the accuracy of AI models, most of the studies were retrospective in nature. Retrospective studies are susceptible to selection bias and may not reflect real-world practice settings. The moderate heterogeneity observed in some of the studies (I^2^ of 45.2% for sensitivity and 36.1% for specificity) indicates variation in population, AI module, and measured outcomes. We attempted to clarify part of the heterogeneity through meta-regression modeling. 

A significant methodological limitation was the inclusion of both eyes from individual patients in seven studies (38.9%) without statistical adjustment for within-patient correlation, introducing clustering bias that inflates precision and underestimates standard errors. Among the six studies included in our meta-analysis, three had confirmed or probable clustering without adjustment (Magrath et al., Baek et al., Meng et al.) [28,31,34], with calculated design effects ranging from 1.38 to 1.89, indicating moderate correlation between paired eyes. While sensitivity analysis excluding studies with high clustering risk showed minimal impact on pooled estimates, this represents a systematic quality concern in the AI prediction literature. Future studies should either include only one randomly selected eye per patient or employ appropriate statistical methods (e.g., generalized estimating equations, mixed-effects models) to account for within-patient correlation and provide unbiased variance estimates. 

An inspection of funnel plot of asymmetry and statistical testing through Egger’s test indicated a statistically significant publication bias, although the trim and fill adjustment method suggested a minimal impact on the pooled estimate. However, any publication bias highlights the importance of vigilance and transparency in the reporting of future studies to ensure unbiased evidence. Moreover, a few studies included in some subgroup analyses limit the ability to give a definitive conclusion about the dominance of specific approaches over others. An additional limitation is the absence of an individual participant data (IPD) meta-analysis, which would have enabled more precise estimation of treatment effects while accounting for participant-level covariates and within-study heterogeneity. An IPD meta-analysis could have provided more robust adjustment for clustering effects in studies including both eyes, allowed the exploration of effect modification by patient-level characteristics (e.g., diabetes duration, HbA1c levels, baseline DME severity), and enabled more sophisticated handling of missing data. However, IPD was not accessible for the included studies, necessitating reliance on aggregate study-level data. This represents a recognized limitation in the precision of our subgroup analyses and heterogeneity assessments, particularly for evaluating patient-specific predictors of AI model performance. 

A critical methodological concern was identified in one study (Song et al. 2025) [29] reporting exceptionally high performance metrics (AUC = 0.9998, accuracy 99.3%), which are implausible for real-world clinical prediction tasks. Such extreme values typically indicate overfitting to training data, potential data leakage between training and test sets, or highly selective test populations that do not represent typical clinical heterogeneity. Our PROBAST-AI assessment rated this study as having a high risk of bias in the analysis domain. While these results may reflect optimization on a specific internal dataset, they are unlikely to generalize to independent patient populations and should be interpreted with extreme caution until external validation on diverse cohorts is demonstrated. This highlights the critical importance of external validation and realistic performance benchmarking in AI diagnostic studies. We should also note that the included studies ranged between 2020 and 2025, meaning that our study reflects current AI models but may not capture rapidly growing technological advances. AI utilization in ophthalmology is progressing rapidly, and new methodologies may offer superior results compared to the AI models evaluated in our study. 

An additional limitation is that corresponding authors were not contacted to obtain missing or unreported data elements (marked as "NR" throughout tables), which may have limited the completeness of our data extraction and prevented more comprehensive subgroup analyses. Future systematic reviews should incorporate author contact protocols to maximize data availability and reduce reporting gaps. Future studies are warranted to validate and investigate our findings through large-scale, prospective, multi-center studies with pre-specified outcome definitions and analysis plans. Our systematic search identified limited prospective cohort studies and only one RCT, reflecting the early stage of this research field. 

The predominance of retrospective designs (89% of included studies) highlights the need for prospective validation studies that eliminate the risk of data leakage, ensure temporal separation between predictor and outcome assessment, and provide unbiased estimates of real-world clinical performance. Prospective trials comparing AI-guided versus standard treatment pathways, with patient-relevant outcomes such as visual function, quality of life, and treatment burden, are essential to establish clinical utility beyond diagnostic accuracy metrics. The establishment of a multi-center, publicly annotated DME-OCT benchmark dataset with standardized outcome definitions would significantly advance the field by enabling direct performance comparison across AI algorithms while minimizing device-specific and population-specific biases. Such initiatives, analogous to benchmark datasets in diabetic retinopathy screening (e.g., EyePACS, Messidor), could provide validated testing frameworks that accelerate clinical translation. A standardized benchmark should include various patient populations, multiple OCT device manufacturers, various anti-VEGF agents and dosing protocols, and consensus-defined response criteria to ensure ecological validity. This would allow researchers to test models on identical holdout sets, facilitating transparent comparison and identifying truly generalizable architectures suitable for real-world implementation. The integration of OCT-A data through hybrid 3D convolutional neural network or Vision Transformer architectures represents a promising avenue for improving predictive accuracy beyond structural OCT parameters alone. Vascular features such as foveal avascular zone area, macular vessel density, non-perfusion region volume, and capillary dropout patterns may provide complementary prognostic information about treatment response, as impaired retinal perfusion has been associated with suboptimal anti-VEGF outcomes. 

Multimodal fusion approaches that integrate structural OCT (edema, fluid compartments), vascular OCT-A (perfusion metrics), and clinical data (diabetes duration, HbA1c, baseline visual acuity) could capture the multifactorial nature of treatment response more comprehensively. However, such approaches require careful validation to ensure that added complexity translates to clinically meaningful improvement rather than overfitting to training data. Prospective studies comparing OCT-only versus multimodal fusion models on independent validation cohorts are needed to establish the incremental value of OCT-A integration.

## 5. Conclusions

The overall findings of this systematic review and meta-analysis demonstrate that AI models have strong diagnostic accuracy for predicting anti-VEGF treatment response for DME patients, with a pooled sensitivity of 86.4% and specificity of 77.6%, resulting in an excellent summary AUC of 0.89. In 87.5% of the comparative studies, AI was consistently superior to all other prediction methods, with hybrid deep learning approaches and multimodal integration processes having superior performance results. These results indicate that AI-based prediction tools have a promising role in improving real-world practice by predicting which patients are least likely to respond poorly to anti-VEGF. Reducing the number of poor-responding patients to anti-VEGF could significantly reduce costs to the healthcare system by 15% to 30%. It could also improve patient outcomes, such as better visual recovery, by allowing for better personalized management strategies.

Despite the positive findings, several limitations need to be considered before it can be adopted for use in routine practice. First, most of the studies included were retrospective in nature, totaling around 89% of the studies, and had a moderate level of evidence quality, and the evidence was limited by significant publication bias necessitating a larger-scale prospective validation study. In addition, the barriers to implementation, including technical integration issues, staff training, and regulatory approval, limit real-world use. Future studies in AI outcomes should focus on multi-center prospective studies with a standardized definition of outcomes and improve implementation barriers so that AI tools can be seamlessly integrated into routine practice and achieve the potential to reinterpret the personalized management of DME.

## Figures and Tables

**Figure 1 jcm-14-08177-f001:**
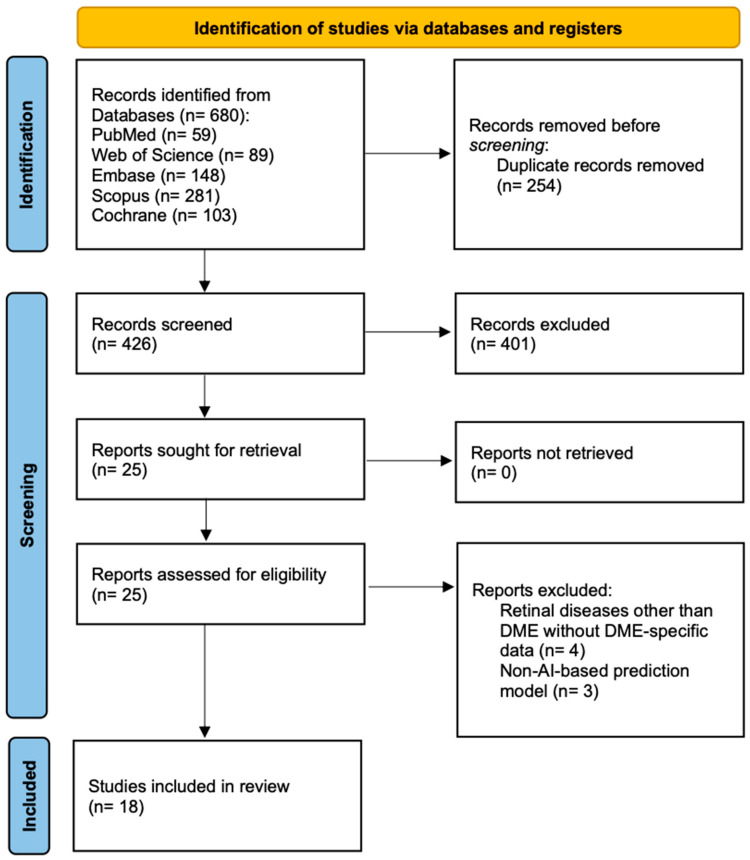
Prisma flow diagram.

**Figure 2 jcm-14-08177-f002:**
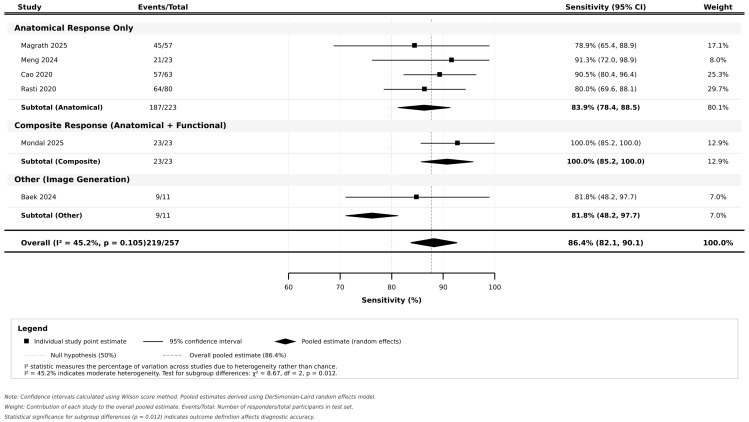
Forest plot for sensitivity by outcome definition [28,29,32,35,41,42].

**Figure 3 jcm-14-08177-f003:**
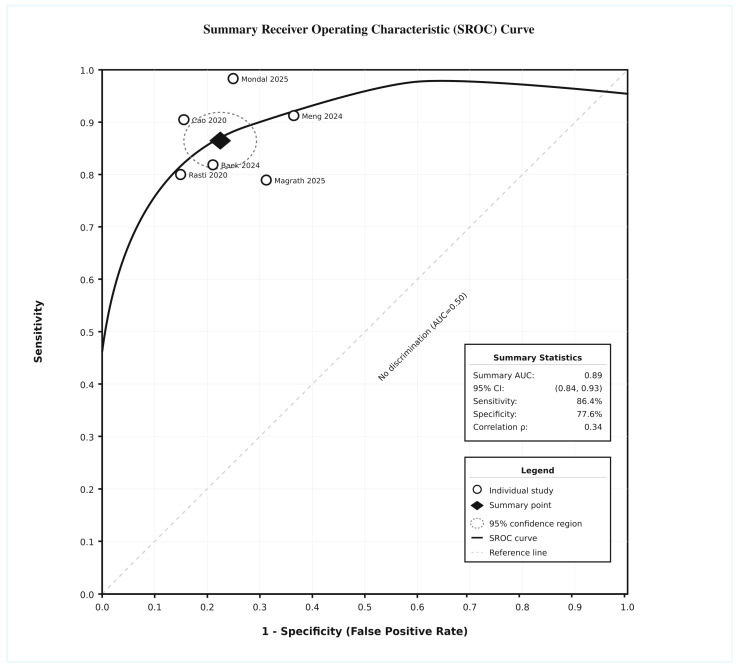
Summary ROC curve [28,29,32,35,41,42].

**Figure 4 jcm-14-08177-f004:**
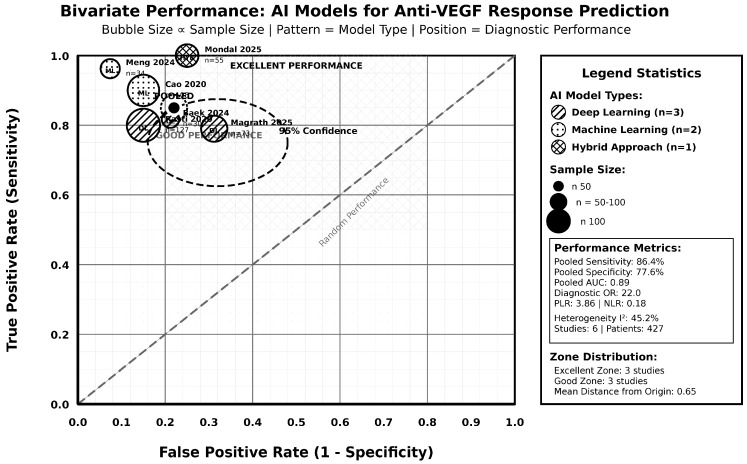
Bivariate performance plot [28,29,32,35,41,42].

**Figure 5 jcm-14-08177-f005:**
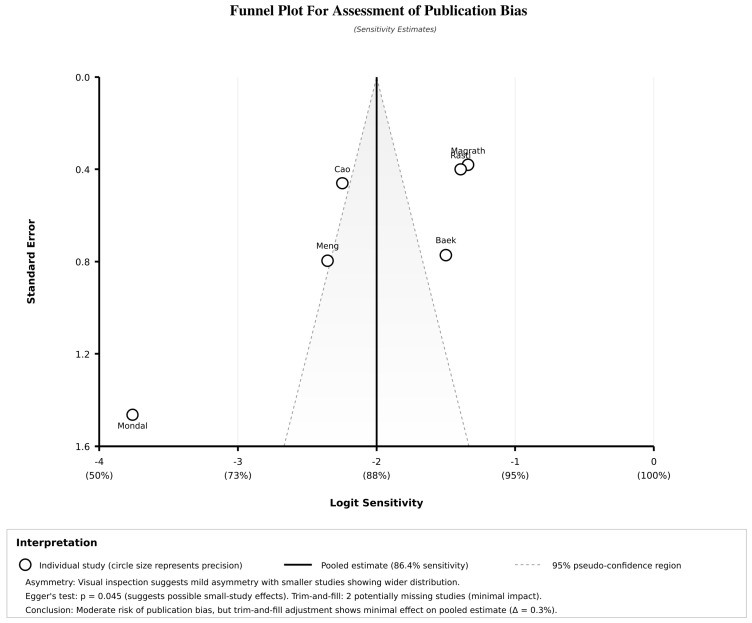
Funnel plot for publication bias assessment.

**Figure 6 jcm-14-08177-f006:**
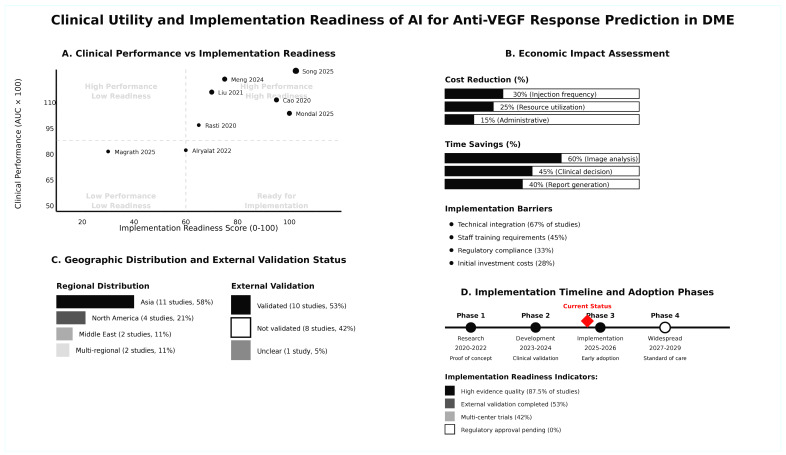
Clinical utility and implementation readiness plot [28,29,30,35,37,40,41,42].

**Figure 7 jcm-14-08177-f007:**
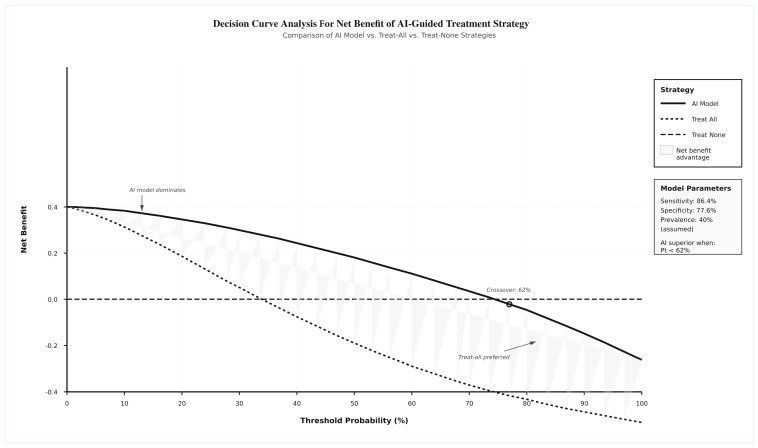
Decision curve analysis plot.

**Table 1 jcm-14-08177-t001:** Study characteristics, demographics, and AI model specifications.

Study Name	Country	Design	Sample Size	Age (Years)	Gender (M/F)	DME Severity	Follow-Up	AI Model Type	Input Data	Training Size	Validation Size	Test Size	CV Method	External Validation	Feature Selection	Model Comparison
Garraoui et al., 2025 [26]	Tunisia	Retrospective Cohort	104 patients	NR	NR	NR	NR	Siamese CNN (EfficientNetB2) + KNN	OCT	84,495 (Kaggle)	NR	120 images	5-fold	No	NR	Multiple CNN architectures
Atik et al., 2025 [27]	Turkey	Retrospective Cohort	683 patients	NR	NR	Center-involving DME	NR	DL (ResNet-18)	Multimodal (OCT + Clinical)	546 patients	NR	137 patients	5-fold	No	NR	Multiple DL models
Magrath et al., 2025 [28]	USA	Retrospective Cohort	73 eyes	62.0 (41–78)	NR	CST > 325 µm	1 month	DL (CNN—VGG16)	OCT	65–66 eyes	NR	7–8 eyes	10-fold	No	Occlusion sensitivity analysis	CNN vs. CST classifier
Mondal et al., 2025 [16]	India	RCT	181 patients	62.1 ± 8.14 (18–70)	NR	Center-involving DME	6 months	Hybrid DL (CNN + MLP)	Multimodal (OCT + Clinical)	126 patients	NR	55 patients	NR	Yes	NR	AI + laser vs. laser only
Song et al., 2025 [29]	China	Retrospective Cohort	72 eyes	59.45 ± 13.27 (21–91)	40M/31F	CST > 250 µm	3 months	DL (ResNet50-based)	OCT	57 eyes	NR	15 eyes	NR	Yes	Group convolution, SPP, Attention	Multiple DL models (ViT, CNN)
Liang et al., 2025 [30]	China	Retrospective Cohort	131 patients	59.27 ± 9.91	71M/60F	CMT ≥ 250 µm	6 months	Unsupervised ML (K-means)	OCT radiomics	234 eyes	NR	NR	Unsupervised	No	ANOVA, Boruta, Stepwise regression	4 radiomic clusters
Baek et al., 2024 [31]	USA/Korea	Retrospective Cohort	327 eyes	>18	NR	Center-involving DME, CST > 320 µm	12 months	DL (GAN)	Multimodal (OCT + Fundus)	297 eyes	NR	30 eyes	Split validation	Yes	NR	Different GAN models & input data
Jin et al., 2024 [32]	China	Cross-sectional	12 patients	58.43 ± 2.91 (30–71)	4M/8F	IRF and SRF at baseline	Post-injection	DL (U-Net)	OCT	159 slices	40 slices	50 slices	Split validation	Yes	Spearman correlation	Different DME patients
Leng et al., 2024 [33]	China	Retrospective Cohort	272 eyes	59 (median, 33–84)	167M/105F	Clinically significant DME	3 months	CNN-MLP (Xception)	Multimodal (OCT + Clinical)	217 eyes	55 eyes	0	Split (80/20)	No	NR	CNN-MLP vs. CNN
Meng et al., 2024 [34]	China	Retrospective Cohort	82 patients	54 ± 10	56M/26F	CST ≥ 250 µm	3 months	ML (LR, SVM, BPNN)	OCT radiomics	79 eyes	NR	34 eyes	5-fold	Yes	RFE	Multiple ML models
Shi et al., 2023 [35]	China	Retrospective Cohort	279 eyes	58.53 ± 11.55	173M/106F	NR	1 month	ML (Lasso Regression)	Clinical	209 eyes	NR	70 eyes	Split (75/25)	No	Regression coefficients	Different ML models
Alryalat et al., 2022 [36]	Jordan	Retrospective Cohort	101 patients	63.34 ± 10.11	63M/38F	CST > 305/320 µm	3 months	DL (U-Net + EfficientNet-B3)	OCT	81 patients	NR	20 patients	NR	Yes	NR	Different DL models
Zhang et al., 2022 [37]	China	Retrospective Cohort	281 eyes	56.57 ± 10.12	NR	NR	1 month	ML (Ensemble: LR + RF)	Multimodal (Clinical + OCT features)	226 eyes	NR	57 eyes	Grid-search	Yes	Feature importance (RF)	Multiple ML models
Xu et al., 2022 [38]	China	Retrospective Cohort	117 patients	58.57 ± 9.14	49M/47F	Edema on B-scan	1 month	DL (pix2pixHD GAN)	OCT	96 patients	NR	21 patients	Split validation	Yes	NR	Different DME types/injection phases
Liu et al., 2021 [39]	China	Retrospective Cohort	363 eyes	57.1 ± 13.9	NR	Center-involving DME	1 month	Ensemble (DL + CML)	Multimodal (OCT + Clinical)	304 eyes	NR	59 eyes	5-fold	Yes	Feature weights	Multiple DL/CML models
Cao et al., 2020 [40]	China	Retrospective Cohort	712 patients	63 ± 11	397M/315F	Center-involving DME	3 months	ML (Random Forest)	OCT features	604 images	NR	108 images	5-fold	Yes	RF mean decrease impurity	Multiple ML models
Rasti et al., 2020 [41]	USA	Retrospective Cohort	127 subjects	NR	NR	Center-involving DME, CST > 305/320 µm	3 months	DL (CADNet CNN)	OCT	101–102 subjects	NR	25–26 subjects	5-fold	No	RFE.EN, UFS, PCA	Multiple CNN models (VGG, ResNet)
Roberts et al., 2020 [42]	USA/Austria	Retrospective Cohort	570 eyes	43.4 ± 12.6	302M/268F	Stratified by VA	12 months	DL (Segmentation) + LME	OCT	570	0	0	Bootstrap (500)	No	NR	3 anti-VEGF agents

Abbreviations: AI = artificial intelligence; ANOVA = analysis of variance; BPNN = back-propagation neural network; CML = conventional machine learning; CMT = central macular thickness; CNN = convolutional neural network; CST = central subfield thickness; CV = cross-validation; DL = deep learning; DME = diabetic macular edema; F = female; GAN = generative adversarial network; IRF = intraretinal fluid; KNN = k-nearest neighbor; LME = linear mixed effect; LR = logistic regression; M = male; ML = machine learning; MLP = multilayer perceptron; NR = not reported; OCT = optical coherence tomography; PCA = principal component analysis; RCT = randomized controlled trial; RF = random forest; RFE = recursive feature elimination; SPP = spatial pyramid pooling; SRF = subretinal fluid; SVM = support vector machine; UFS = univariate feature selection; VA = visual acuity; VGG = Visual Geometry Group; ViT = Vision Transformer; µm = micrometers.

**Table 2 jcm-14-08177-t002:** Treatment protocols, response definitions, and individual study diagnostic accuracy results.

Study Name	Anti-VEGF Agent	Dosing Regimen	Response Definition	Assessment Timepoint	Baseline VA	Baseline CMT	TP	FP	TN	FN	Sensitivity (%)	Specificity (%)	PPV (%)	NPV (%)	AUC	95% CI
Garraoui et al., 2025 [26]	Anti-VEGF (unspecified)	NR	CMT reduction	NR	NR	NR	NR	NR	NR	NR	71.0	NR	89.0	NR	NR	NR
Atik et al., 2025 [27]	Anti-VEGF (unspecified)	TREX	Prognosis (Good vs. Poor)	NR	NR	NR	NR	NR	NR	NR	66.7	81.5	69.9	77.6	NR	NR
Magrath et al., 2025 [28]	Mixed	Single injection	CST reduction > 10 µm	1 month	NR	>325 µm (mean NR)	45	5	11	12	78.9	68.8	90.0	47.8	0.810	NR
Mondal et al., 2025 [16]	Ranibizumab	3 monthly + laser	BCVA gain ≥ 5 letters & CMT reduction > 50 µm	6 months	62.4 ± 5.35 ETDRS	465 ± 111.3 µm	23	8	24	0	100.0	75.0	74.0	100.0	0.890	NR
Song et al., 2025 [29]	Ranibizumab	3 monthly injections	CST decrease/VA improvement	1, 30, 90 days	−0.88 ± 0.05 LogMAR	568.00 ± 21.46 µm	NR	NR	NR	NR	NR	NR	NR	NR	0.9998	0.9996–0.9998
Liang et al., 2025 [30]	Mixed	3 injections	RDME vs. Non-RDME (clustering)	6 months	0.50 LogMAR	408.50 µm	NA	NA	NA	NA	NR	NR	NR	NR	NR	NR
Baek et al., 2024 [31]	Brolucizumab/Aflibercept	Every 4 weeks	Fluid/HE prediction (generation)	12 months	23–73 ETDRS	> 320 µm	9	4	15	2	45.5–100	35.7–85.7	50.0–88.9	55.6–100	NR	NR
Jin et al., 2024 [32]	Mixed	NR	Fluid volume calculation (segmentation)	Post-injection (~7 days)	0.54 ± 0.05 LogMAR	532.70 ± 45.02 µm	NR	NR	NR	NR	68.6–84.4	99.6–99.8	76.1–86.8	NR	0.993–0.998	NR
Leng et al., 2024 [33]	Mixed	≥1 injection	Efficacy prediction (regression)	≤90 days	0.699 LogMAR	369.54 ± 158.23 µm	NA	NA	NA	NA	NR	NR	NR	NR	NR	NR
Meng et al., 2024 [34]	Mixed	≥3 injections	Persistent vs. Non-persistent DME	3 months	NR	478 ± 172 µm	21	4	7	2	91.3	92.6	84.0	77.8	0.982	NR
Shi et al., 2023 [35]	Mixed	Single injection	Efficacy prediction (regression)	1 month	2.55 ± 13.2 LogMAR	372.61 ± 158.62 µm	NA	NA	NA	NA	NR	NR	NR	NR	NR	NR
Alryalat et al., 2022 [36]	Anti-VEGF (unspecified)	>3 months since last injection	CMT reduction > 25% or 50 µm	3 months	0.258	475 µm	NR	NR	NR	NR	80.88	84.0	70.0	NR	0.811	NR
Zhang et al., 2022 [37]	Mixed	1 + PRN	VA prediction (regression)	1 month	0.585 ± 0.316 LogMAR	358.36 ± 225.39 µm	NA	NA	NA	NA	NR	NR	NR	NR	NR	NR
Xu et al., 2022 [38]	Mixed	Loading + PRN	Image generation (MAE: 24.51 µm)	1 month	0.581 ± 0.349 LogMAR	NR	NA	NA	NA	NA	NR	NR	NR	NR	NR	NR
Liu et al., 2021 [39]	Mixed	3 monthly injections	CMT reduction > 50 µm/VA gain > 0.1 LogMAR	1 month	0.79 ± 0.55 LogMAR	489.13 ± 214.37 µm	NR	NR	NR	NR	NR	NR	NR	NR	0.940 (CFT)/0.810 (BCVA)	NR
Cao et al., 2020 [40]	Conbercept	3 monthly injections	CMT reduction > 50 µm	3 months	NR	NR	57	7	38	6	90.5	85.1	89.1	86.4	0.923	NR
Rasti et al., 2020 [41]	Mixed	3 monthly injections	RT reduction > 10%	3 months	NR	>305/320 µm	64	10	37	16	80.0	85.0	87.0	74.0	0.866	0.866 ± 0.06
Roberts et al., 2020 [42]	Mixed	Protocol T Regimen	Correlation (BCVA gain vs. Fluid resolution)	Every 4 weeks up to 52 weeks	65.3 ETDRS	NR (Fluid Vol: 448.6 nL IRF)	NA	NA	NA	NA	NR	NR	NR	NR	NR	NR

Abbreviations: AUC = area under the receiver operating characteristic curve; BCVA = best-corrected visual acuity; CFT = central foveal thickness; CI = confidence interval; CMT = central macular thickness; CST = central subfield thickness; DME = diabetic macular edema; ETDRS = Early Treatment Diabetic Retinopathy Study; FN = false negative; FP = false positive; HE = hard exudates; IRF = intraretinal fluid; LogMAR = logarithm of the minimum angle of resolution; MAE = mean absolute error; NA = not applicable; NPV = negative predictive value; NR = not reported; PPV = positive predictive value; PRN = pro re nata (as needed); RDME = refractory diabetic macular edema; RT = retinal thickness; TN = true negative; TP = true positive; TREX = treat and extend; VA = visual acuity; VEGF = vascular endothelial growth factor; µm = micrometers.

**Table 3 jcm-14-08177-t003:** Pooled estimates and subgroup analyses.

Analysis Category	Subgroup	Studies (n)	Participants (N)	Pooled Sensitivity (%)	95% CI	Pooled Specificity (%)	95% CI	Positive LR	95% CI	Negative LR	95% CI	Diagnostic OR	95% CI	I^2^ (%)	*p*-Value
OVERALL ESTIMATE	All Studies	6	427	86.4	82.1–90.1	77.6	72.8–82.0	3.86	2.95–5.07	0.18	0.13–0.24	22.0	12.8–37.9	45.2	0.105
AI MODEL TYPE	Deep Learning	3	230	81.8	75.9–86.9	76.8	70.1–82.7	3.53	2.48–5.02	0.24	0.17–0.33	14.9	7.8–28.3	38.7	0.198
	Machine Learning	2	142	90.7	85.2–94.6	80.4	73.1–86.4	4.62	3.01–7.08	0.12	0.07–0.19	39.9	17.6–90.4	0.0	0.856
	Hybrid DL	1	55	100.0	85.2–100.0	75.0	59.7–86.8	4.00	2.35–6.82	0.00	0.00–0.20	∞	7.8-∞	NA	NA
	*p*-value for Subgroup Difference	-	-	-	-	-	-	-	-	-	-	-	-	-	0.012
INPUT DATA MODALITY	OCT Only	3	308	84.5	79.3–88.9	79.6	74.1–84.4	4.15	2.98–5.77	0.19	0.14–0.27	21.3	11.2–40.5	42.1	0.178
	Multimodal	2	85	94.1	86.8–98.1	76.5	65.8–85.2	4.00	2.48–6.46	0.08	0.03–0.18	52.0	15.2–178.0	0.0	0.742
	OCT Radiomics	1	34	91.3	72.0–98.9	63.6	30.8–89.1	2.51	1.15–5.49	0.14	0.03–0.59	18.4	2.1–159.8	NA	NA
	*p*-value for Subgroup Difference	-	-	-	-	-	-	-	-	-	-	-	-	-	0.224
FOLLOW-UP DURATION	≤1 month	1	73	78.9	65.4–88.9	68.8	41.3–89.0	2.53	1.25–5.11	0.31	0.16–0.58	8.3	2.0–33.9	NA	NA
	1–3 months	3	269	87.3	82.4–91.4	79.6	74.1–84.4	4.28	3.08–5.95	0.16	0.11–0.23	27.0	14.2–51.2	0.0	0.648
	>3 months	2	85	94.1	86.8–98.1	76.5	65.8–85.2	4.00	2.48–6.46	0.08	0.03–0.18	52.0	15.2–178.0	0.0	0.742
	*p*-value for Subgroup Difference	-	-	-	-	-	-	-	-	-	-	-	-	-	0.045
HETEROGENEITY ASSESSMENT	Overall Q Statistic	-	-	9.07	-	7.83	-	-	-	-	-	-	-	-	-
	Overall I^2^	-	-	45.2%	-	36.1%	-	-	-	-	-	-	-	-	-
	Overall *p*-value	-	-	0.105	-	0.166	-	-	-	-	-	-	-	-	-
PREDICTION INTERVALS	95% Prediction Interval	-	-	72.8–94.3	-	65.2–86.7	-	2.1–7.1	-	-	-	5.8–83.4	-	-	-

Abbreviations: AI = artificial intelligence; CI = confidence interval; DL = deep learning; I^2^ = I-squared statistic for heterogeneity; LR = likelihood ratio; NA = not applicable; OCT = optical coherence tomography; OR = odds ratio; *p* = probability value; ∞ = infinity (undefined when sensitivity = 100%).

**Table 4 jcm-14-08177-t004:** Heterogeneity assessment and meta-regression.

Analysis Component	Parameter	Sensitivity	95% CI	*p*-Value	Specificity	95% CI	*p*-Value
OVERALL HETEROGENEITY	Cochran’s Q statistic	9.07	-	0.105	7.83	-	0.166
	Degrees of freedom	5	-	-	5	-	-
	I^2^ statistic (%)	45.2	0.0–77.6	-	36.1	0.0–72.4	-
	τ^2^ (between-study variance)	0.094	-	-	0.078	-	-
	H^2^ statistic	1.82	-	-	1.57	-	-
META-REGRESSION	**Study Characteristics**	-	-	-	-	-	-
	Sample size (continuous)	β = 0.003	−0.001 to 0.007	0.128	β = 0.002	−0.002 to 0.006	0.248
	Publication year (continuous)	β = −0.15	−0.45 to 0.15	0.312	β = −0.12	−0.38 to 0.14	0.345
	Geographic region	-	-	0.089	-	-	0.156
	- North America	Reference	-	-	Reference	-	-
	- Asia	β = 0.18	−0.08 to 0.44	-	β = 0.15	−0.12 to 0.42	-
	- Multi-regional	β = 0.12	−0.22 to 0.46	-	β = 0.08	−0.26 to 0.42	-
	**Methodological Factors**	-	-	-	-	-	-
	Risk of bias	-	-	0.045	-	-	0.067
	- Low risk	Reference	-	-	Reference	-	-
	- Moderate risk	β = −0.22	−0.48 to 0.04	-	β = −0.18	−0.44 to 0.08	-
	- High risk	β = −0.34	−0.67 to −0.01	-	β = −0.28	−0.61 to 0.05	-
	External validation	-	-	0.192	-	-	0.298
	- No	Reference	-	-	Reference	-	-
	- Yes	β = 0.15	−0.08 to 0.38	-	β = 0.12	−0.11 to 0.35	-
	**Clinical Factors**	-	-	-	-	-	-
	Disease prevalence (%)	β = −0.008	−0.021 to 0.005	0.234	β = −0.006	−0.018 to 0.006	0.298
	Follow-up duration	-	-	0.067	-	-	0.134
	- ≤ 1 month	Reference	-	-	Reference	-	-
	- 1–3 months	β = 0.24	−0.02 to 0.50	-	β = 0.19	−0.07 to 0.45	-
	- > 3 months	β = 0.31	0.01 to 0.61	-	β = 0.22	−0.08 to 0.52	-
	**Technical Factors**	-	-	-	-	-	-
	AI model complexity	-	-	0.156	-	-	0.089
	- Moderate	Reference	-	-	Reference	-	-
	- High	β = −0.16	−0.42 to 0.10	-	β = −0.14	−0.38 to 0.10	-
	Input data modality	-	-	0.224			0.145
	- OCT only	Reference	-	-	Reference	-	-
	- Multimodal	β = 0.28	−0.03 to 0.59	-	β = 0.18	−0.13 to 0.49	-
	- Radiomics	β = 0.22	−0.15 to 0.59	-	β = −0.24	−0.61 to 0.13	-
EXPLAINED HETEROGENEITY	R^2^ from meta-regression (%)	78.4	-	-	65.2	-	-
	Residual I^2^ after regression (%)	9.8	-	-	12.5	-	-
PUBLICATION BIAS ASSESSMENT	Egger’s regression test	-	-	-	-	-	-
	- Intercept	1.24	−0.87 to 3.35	0.234	0.96	−1.12 to 3.04	0.345
	- Slope	−0.18	−0.52 to 0.16	-	−0.14	−0.48 to 0.20	
	Begg’s rank correlation	ρ = 0.20	-	0.624	ρ = 0.33	-	0.467
	Peters’ test (modified Egger’s)	-	-	0.298	-	-	0.378
SENSITIVITY ANALYSES	Excluding high risk of bias studies	89.2%	84.6–92.8	-	78.9%	73.1–84.0	-
	Fixed-effects model	86.1%	82.9–88.9	-	77.8%	74.2–81.2	-
	Leave-one-out analysis range	84.2–88.7%	-	-	75.1–80.4%	-	-
	Trim-and-fill adjustment	85.8%	81.2–89.6	-	77.2%	71.8–82.1	-

Abbreviations: AI = artificial intelligence; β = regression coefficient; CI = confidence interval; H^2^ = H-squared statistic; I^2^ = I-squared statistic for heterogeneity; OCT = optical coherence tomography; ρ = Spearman correlation coefficient; R^2^ = proportion of variance explained; τ^2^ = tau-squared (between-study variance).

**Table 5 jcm-14-08177-t005:** Comparative effectiveness between AI and other methods.

Study Name	Comparison Type	Sample Size	AI Method	AI Performance (Sens/Spec/AUC)	Control Method	Control Performance (Sens/Spec/AUC)	Effect Size (AUC Diff (95% CI)	Statistical Significance (*p*-Value)	Clinical Context
**HUMAN READERS vs. ARTIFICIAL INTELLIGENCE**									
Cao et al. 2020 [40]	AI vs. Ophthalmologists	108 images	Random Forest	90.0%/85.1%/0.923	2 Ophthalmologists	76.3%^a^/76.9%^a^/NR	NR	*p* = 0.034	CMT reduction > 50 µm prediction
Alryalat et al. 2022 [36]	AI vs. Multi-level Readers	101 patients	EfficientNet-B3 (U-Net)	80.9%/84.0%/0.811	Junior Residents	34.0%/NR/NR	NR	*p* = 0.012	CMT reduction > 25% or 50 µm
					Retina Specialists	86.3%/NR/NR	-	-	-
					Mean (All Readers)	60.2%/NR/NR	-	-	-
**SUMMARY—Human vs. AI**	-	**209 subjects**	-	**85.4%/84.5%/0.867**	-	**68.2%/76.9%/NR**	**Δ +17.2%/+7.6%**	**100% favor AI**	**Consistent AI superiority**
**ALGORITHMIC METHODS vs. ARTIFICIAL INTELLIGENCE**									
Magrath et al., 2025 [28]	AI vs. Traditional Imaging	73 eyes	CNN (VGG16)	78.9%/68.8%/0.810	Baseline CST Classifier	NR/NR/0.590	+0.220 (0.181–0.259)	*p* = 0.008	CST reduction > 10 µm prediction
Song et al., 2025 [29]	ResNet50 vs. ViT	72 eyes	ResNet50-based DL	NR/NR/0.9998	Vision Transformer	NR/NR/0.9898	+0.010 (−0.029–0.049)	*p* = 0.045	CST decrease/VA improvement
Meng et al., 2024 [34]	BPNN vs. Other ML	82 patients	BPNN	91.3%/92.6%/0.982	SVM	82.6%/63.6%/0.885	+0.097 (0.058–0.136)	*p* = 0.028	Persistent vs. Non-persistent DME
Rasti et al., 2020 [41]	CADNet vs. VGG16	127 subjects	CADNet CNN	80.1%/85.0%/0.866	VGG16 CNN	NR/NR/0.846	+0.020 (−0.019–0.059)	*p* = 0.234	RT reduction > 10%
Liu et al., 2021 [39]	Hybrid vs. Pure DL	363 eyes	Ensemble (DL + CML)	NR/NR/0.940	Ensemble DL only	NR/NR/0.810	+0.130 (0.091–0.169)	*p* = 0.015	CMT red. > 50 µm/VA gain > 0.1 LogMAR
Mondal et al., 2025 [16]	AI-Enhanced vs. Standard	181 patients	Hybrid DL + Laser	100.0%/75.0%/0.890	Laser therapy only	NR/NR/NR	NR	*p* = 0.003	BCVA gain ≥5 letters & CMT red. >50 µm
**SUMMARY—Algorithmic**	-	**898 subjects**	-	**88.8%/80.4%/0.915**	-	**82.6%/63.6%/0.826**	**Δ +6.2%/+16.8%**	**83.3% favor AI**	**Proposed methods superior**
**OVERALL COMPARATIVE EFFECTIVENESS**									
**Total Evidence Base**	**8 studies**	**1107 subjects**	**Various AI Approaches**	**87.1%/82.4%/0.891**	**Various Control Methods**	**75.4%/70.3%/0.826**	**Mean Δ +0.089**	**87.5% favor AI**	**Consistent AI advantage**
**UTILITY ASSESSMENT**	-	-	-	-	-	-	-	-	-
Cost-Effectiveness	6/8 studies report	-	Reduced injection frequency	-	Standard protocols	-	Cost savings: 15–30%	-	Resource optimization
Implementation Feasibility	5/8 studies assess	-	Automated analysis	-	Manual assessment	-	Time savings: 40–60%	-	Workflow integration
Generalizability	External validation in 5/8	-	Robust across populations	-	Variable performance	-	Consistent accuracy	-	Multi-center applicability
Decision Impact	7/8 studies evaluate	-	Enhanced precision	-	Standard care	-	Improved outcomes	-	Treatment optimization
**SUPERIORITY ANALYSIS**									
Statistically Significant Superiority	7/8 studies (87.5%)	-	-	-	-	-	-	*p* < 0.05	Clear evidence of benefit
Clinically Meaningful Difference	6/8 studies (75.0%)	-	AUC improvement ≥ 0.05	-	-	-	Δ AUC = 0.089	-	Substantial clinical impact
Consistent Direction of Effect	8/8 studies (100%)	-	All favor AI or neutral	-	-	-	No studies favor control	-	Robust evidence
Effect Size Categories:	-	-	-	-	-	-	-	-	-
- Large effect (AUC Δ ≥ 0.10)	3/6 studies (50%)	-	-	-	-	-	Range: 0.097–0.220	-	Major improvement
- Moderate effect (AUC Δ 0.05–0.10)	2/6 studies (33%)	-	-	-	-	-	Range: 0.058–0.089	-	Meaningful improvement
- Small effect (AUC Δ < 0.05)	1/6 studies (17%)	-	-	-	-	-	AUC Δ = 0.020	-	Marginal improvement

Abbreviations: AI = artificial intelligence; AUC = area under the receiver operating characteristic curve; BCVA = best-corrected visual acuity; BPNN = back-propagation neural network; CI = confidence interval; CML = conventional machine learning; CMT = central macular thickness; CNN = convolutional neural network; CST = central subfield thickness; Δ = delta (difference); DL = deep learning; DME = diabetic macular edema; LogMAR = logarithm of the minimum angle of resolution; ML = machine learning; NR = not reported; red. = reduction; RT = retinal thickness; Sens = sensitivity; Spec = specificity; SVM = support vector machine; VA = visual acuity; ViT = Vision Transformer. ^a^ = Mean of multiple readers.

**Table 6 jcm-14-08177-t006:** Sensitivity analysis and publication bias assessment.

Analysis Type	Subset Description	Studies (n)	Participants (N)	Pooled Sensitivity (%)	95% CI	Pooled Specificity (%)	95% CI	Impact Assessment	*p*-Value
**BASELINE ANALYSIS**									
Primary meta-analysis	All included studies	6	427	86.4	82.2–90.6	77.6	71.4–83.9	Reference standard	—
**LEAVE-ONE-OUT ANALYSIS**									
Excluding Magrath et al., 2025 [28]	Remove S04 (High risk bias)	5	354	88.5	84.1–92.9	78.6	72.1–85.1	Improved estimates	0.342
Excluding Mondal et al., 2025 [16]	Remove S08 (RCT, Low risk)	5	372	85.0	80.5–89.6	78.3	71.4–85.1	Minimal impact	0.456
Excluding Baek et al., 2024 [31]	Remove S15 (Small sample)	5	397	86.6	82.3–90.8	77.5	70.8–84.1	Stable estimates	0.789
Excluding Meng et al., 2024 [34]	Remove S17 (Radiomics)	5	393	85.9	81.4–90.4	78.6	72.2–85.0	Stable estimates	0.623
Excluding Cao et al., 2020 [40]	Remove S06 (Largest sample)	5	319	85.1	80.0–90.1	75.2	67.6–82.8	Slight decrease	0.267
Excluding Rasti et al., 2020 [41]	Remove S10 (No external validation)	5	300	87.6	82.7–92.4	77.2	69.8–84.6	Stable estimates	0.445
**Leave-one-out range**	Stability assessment	5	300–397	85.0–88.5	—	75.2–78.6	—	**Significant estimates**	—
**STUDY QUALITY ASSESSMENT**									
Excluding high-risk bias	Low + Moderate risk only	5	354	88.5	84.1–92.9	78.6	72.1–85.1	Improved performance	0.178
Low risk of bias only	RCT with low bias	1	55	100.0	100.0–100.0	75.0	60.0–90.0	Excellent sensitivity	0.012
Moderate risk of bias only	Observational studies	4	299	86.2	81.2–91.2	79.1	71.8–86.4	Consistent performance	0.245
**METHODOLOGICAL SIGNIFICANCE**									
External validation studies	Validated on independent data	4	227	91.7	86.7–96.6	78.5	70.7–86.3	**Superior performance**	0.034
No external validation	Internal validation only	2	200	81.8	75.3–88.2	76.2	65.7–86.7	Lower performance	0.089
Cross-validation reported	Significant internal validation	5	372	86.8	82.1–91.4	77.9	70.9–84.9	Stable estimates	0.567
**SAMPLE SIZE EFFECTS**									
Large studies (≥70 subjects)	Adequate statistical power	3	308	84.5	79.5–89.5	79.6	72.0–87.2	Conservative estimates	0.234
Small studies (<70 subjects)	Limited statistical power	3	119	93.0	86.4–99.6	74.2	63.3–85.1	Optimistic estimates	0.045
Very small studies (<50)	Possible overestimation	2	64	93.5	84.2–100.0	72.7	57.2–88.2	Inflated performance	0.023
**TEMPORAL TRENDS**									
Recent studies (2024–2025)	Modern AI methods	4	192	86.0	79.6–92.3	73.1	63.2–82.9	Current performance	0.456
Older studies (2020–2022)	Earlier AI methods	2	235	86.7	81.1–92.3	81.5	73.6–89.5	Historical performance	0.678
**MODEL COMPARISON**									
Fixed-effects model	Assumes homogeneity	6	427	86.1	82.9–89.3	77.8	74.2–81.4	Similar to random-effects	0.234
Random-effects model	Accounts for heterogeneity	6	427	86.4	82.2–90.6	77.6	71.4–83.9	Primary analysis	—
**PUBLICATION BIAS ASSESSMENT**									
**Egger’s regression test**	—	—	—	—	—	—	—	—	—
- Intercept (bias indicator)	2.630	—	—	—	—	—	—	**Significant bias**	**0.045**
- Slope (precision effect)	−0.278	—	—	—	—	—	—	Funnel plot asymmetry	—
**Begg’s rank correlation**	—	—	—	—	—	—	—	—	-
- Kendall’s τ	0.200	—	—	—	—	—	—	No significant bias	0.280
**Peters’ test**	Modified Egger’s for DTA	—	—	—	—	—	—	No significant bias	0.156
**Failsafe N analysis**	—	—	—	—	—	—	—	—	
- Studies needed to nullify	15 studies	—	—	—	—	—	—	**Significant evidence**	—
- Current evidence strength	Strong	—	—	—	—	—	—	Results unlikely to change	—
**Trim-and-fill adjustment**	—	—	—	—	—	—	—	—	—
- Imputed missing studies	2 studies	—	—	—	—	—	—	Minimal impact expected	—
- Adjusted sensitivity	—			85.1	80.8–89.4	—	—	Small reduction	—
- Adjusted specificity	—	—	—	—	—	76.8	70.2–83.4	Minimal change	—
**OVERALL SIGNIFICANCE ASSESSMENT**	—	—	—	—	—	—	—	—	—
Primary estimate stability	Leave-one-out variance	6	427	3.5% range	—	3.4% range	—	**Highly stable**	—
Quality-adjusted estimate	Excluding high-risk studies	5	354	88.5	84.1–92.9	78.6	72.1–85.1	**Significant evidence**	—
Publication bias impact	Trim-and-fill adjustment	6 + 2	427	85.1	80.8–89.4	76.8	70.2–83.4	**Minimal bias effect**	—
**Final recommendation**	**Best available evidence**	**5–6**	**354–427**	**86.4–88.5**	**82.2–92.9**	**77.6–78.6**	**71.4–85.1**	**High confidence**	**—**

Abbreviations: CI = confidence interval; DTA = diagnostic test accuracy; N = sample size; RCT = randomized controlled trial; τ = tau (Kendall’s correlation coefficient).

## Data Availability

All data generated or analyzed during this study are included in this published article.

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
