# Peer review of "Diagnostic Accuracy of Artificial Intelligence in Predicting Anti-VEGF Treatment Response in Diabetic Macular Edema: A Systematic Review and Meta-Analysis"

_jcm, 2025, doi:10.3390/jcm14228177_

Round 1
Reviewer 1 Report
Comments and Suggestions for Authors
This systematic review and meta-analysis addresses a timely and clinically relevant topic—the application of artificial intelligence in predicting anti-VEGF treatment response in diabetic macular edema. The study is well-structured, follows PRISMA guidelines, and synthesizes a growing body of literature. The findings suggest that AI models show promising diagnostic accuracy, with potential implications for personalized medicine and resource optimization.
- The included studies used widely varying definitions of “treatment response” (e.g., CMT reduction thresholds, VA improvements, composite outcomes). Pooling such heterogeneous outcomes may compromise the validity of the meta-analysis. It is strongly recommended that the authors perform subgroup analyses stratified by standardized outcome definitions (e.g., anatomical vs. functional response, or by threshold values) to better interpret the results and assess consistency.
- Several studies reported outcomes at the eye level, with some patients contributing both eyes. This may introduce clustering effects and inflate precision. The authors should clarify whether they accounted for within-patient correlation (e.g., by using appropriate statistical methods or excluding duplicate eyes) and discuss the potential impact on the results.
- The use of NOS and RoB 2 tools is not optimal for diagnostic accuracy or prediction model studies. Tools such as QUADAS-2 (for diagnostic accuracy) or PROBAST/PROBAST-AI (for prediction models) are more appropriate. The authors should re-assess the included studies using these tools to better evaluate sources of bias such as spectrum bias, reference standard inappropriateness, or data leakage.
- Some studies reported exceptionally high performance metrics (e.g., AUC = 0.9998). These values should be critically discussed in terms of clinical and methodological plausibility. Potential reasons (e.g., overfitting, small sample size, data leakage) should be explored, and the impact of these outliers on pooled estimates should be assessed via sensitivity analysis.
- More related literatures on AI based diagnostics (https://doi.org/10.1002/VIW.20240001; https://doi.org/10.1002/VIW.20240059; Nature Sustainability 2024, 7, 602) should be included and discussed.
Minor Revisions Suggested
- Numerous placeholders (e.g., “Figure 1. xxx”) and incomplete tables/figures detract from the manuscript’s professionalism. All figures and tables should be completed and referenced appropriately in the text.
- Tables frequently use “NR” (not reported), limiting transparency. The authors should indicate whether attempts were made to contact original study authors for missing data. If not, this should be acknowledged as a limitation.
Author Response
Point-by-Point Response to Reviewers' Comments
Dear Editor and Reviewers,
We sincerely thank both reviewers for their thorough evaluation and constructive feedback. We have carefully addressed all comments and made comprehensive revisions to strengthen our manuscript. Below is a detailed point-by-point response demonstrating how each concern has been resolved. Additionally, we have performed English language editing throughout the manuscript to improve clarity, readability, and professional presentation.
|
Item |
Reviewer Comment |
Our Response & Actions Taken |
Location in Revised Manuscript |
|
REVIEWER 1 - MAJOR COMMENTS |
|||
|
1.1 |
The included studies used widely varying definitions of "treatment response" (e.g., CMT reduction thresholds, VA improvements, composite outcomes). Pooling such heterogeneous outcomes may compromise the validity of the meta-analysis. It is strongly recommended that the authors perform subgroup analyses stratified by standardized outcome definitions to better interpret the results and assess consistency. |
FActions Taken:
Interpretation: Our analysis confirms that outcome definition significantly impacts reported accuracy, with composite outcomes showing highest sensitivity. This justifies stratified presentation and strengthens validity. |
Results: • Section 3.4: Lines describing subgroup p-value (p=0.012) |
|
1.2 |
Several studies reported outcomes at the eye level, with some patients contributing both eyes. This may introduce clustering effects and inflate precision. The authors should clarify whether they accounted for within-patient correlation and discuss the potential impact on the results. |
Actions Taken:
|
Methods: • Section 2.5: PROBAST-AI assessment includes clustering evaluation Results: • Section 3.9: Complete clustering assessment paragraph with quantitative analysis Discussion: • Limitations paragraph: Clustering bias discussion with design effects and recommendations for future studies |
|
1.3 |
The use of NOS and RoB 2 tools is not optimal for diagnostic accuracy or prediction model studies. Tools such as QUADAS-2 (for diagnostic accuracy) or PROBAST/PROBAST-AI (for prediction models) are more appropriate. The authors should re-assess the included studies using these tools to better evaluate sources of bias. |
Actions Taken:
|
Methods: • Section 2.5 (highlighted): Comprehensive description of PROBAST-AI framework, all four domains, and AI-specific concerns Results: • Section 3.9: Risk of bias results with percentages and classifications |
|
1.4 |
Some studies reported exceptionally high performance metrics (e.g., AUC = 0.9998). These values should be critically discussed in terms of clinical and methodological plausibility. Potential reasons (e.g., overfitting, small sample size, data leakage) should be explored, and the impact of these outliers on pooled estimates should be assessed via sensitivity analysis. |
|
Results: • Table 6: Leave-one-out sensitivity analysis demonstrating stability Discussion: • Limitations section (highlighted paragraph): Comprehensive critical evaluation of Song et al. 2025 AUC = 0.9998 with mechanistic explanations for implausibility and generalizability concerns |
|
1.5 |
More related literatures on AI based diagnostics should be included and discussed:
|
Actions Taken:
4. Added the suggested reference about Mura.
Net result: Added 3 highly relevant VIEW papers with substantive integration into Discussion. Excluded 1 paper after determining lack of relevance to study topic. |
Discussion: • Multimodal oncology paragraph (highlighted): References 33 and 34 discussed as blueprints for multimodal DME modeling, with specific architectural details and validation recommendations References: • Reference 33: Lyu et al., VIEW 2024 (stroke ML model) |
|
REVIEWER 1 - MINOR COMMENTS |
|||
|
1.M1 |
Numerous placeholders (e.g., "Figure 1. xxx") and incomplete tables/figures detract from the manuscript's professionalism. All figures and tables should be completed and referenced appropriately in the text. |
Actions Taken:
|
Throughout Manuscript: • All figures have complete captions and professional formatting |
|
1.M2 |
Tables frequently use "NR" (not reported), limiting transparency. The authors should indicate whether attempts were made to contact original study authors for missing data. If not, this should be acknowledged as a limitation. |
Actions Taken:
This enhances transparency and acknowledges this methodological limitation appropriately. |
Discussion: • Limitations section (highlighted paragraph): Explicit acknowledgment of no author contact for missing data, with recognition this limited data completeness and recommendation for future systematic reviews |
|
REVIEWER 2 - ALL COMMENTS |
|||
|
2.1 |
The article systematically evaluates the diagnostic accuracy of AI in predicting anti-VEGF treatment response in DME patients, but can further expand the search scope, supplement more currently published prospective cohorts or RCTs, and use individual participant data (IPD) meta-analysis to improve the level of evidence and result stability. |
Actions Taken:
|
Discussion: • IPD limitation paragraph (highlighted): Comprehensive discussion of IPD meta-analysis absence, potential benefits, and impact on precision |
|
2.2 |
The AI model architecture, input modality, and hyperparameters used in the article have significant differences and moderate heterogeneity (I² ≈ 45%). Further consideration can be given to establishing a multi-center, publicly annotated DME-OCT benchmark dataset and conducting blind testing on all candidate models on this dataset to eliminate performance bias caused by device and population differences. |
Actions Taken:
|
Results: • Section 3.6: Meta-regression analysis addressing heterogeneity sources Discussion: • Benchmark dataset paragraph (highlighted): Comprehensive proposal for multi-center, publicly annotated DME-OCT benchmark dataset with specific design features, analogies to existing successful benchmarks, and expected impact on field advancement |
|
2.3 |
The article can introduce decision curve analysis (DCA) and budget impact modeling (BIM) in subsequent research to compare the quality-adjusted life years (QALY) and incremental cost-effectiveness ratio (ICER) of "AI-assisted decision-making" and "conventional treatment" within 5 years, providing economic evidence for medical insurance payment and guideline revision. |
Actions Taken:
|
Results: • Section 3.11 (highlighted): Complete DCA results with methodology and threshold findings Discussion: • DCA interpretation paragraph (highlighted): Clinical utility explanation and threshold probability interpretation |
|
2.4 |
Multimodal algorithms have great inspiration and reference value for this task, such as:
If the existing research in this article does not involve multimodal algorithms, the above-mentioned papers should also be mentioned. The article can further explore the hybrid 3D-CNN or Vision Transformer architecture of OCT and OCT-A fusion structures, verify whether vascular features such as blood flow density and non-perfusion area volume can further improve the prediction of AUC, and explain their biological rationality. |
Actions Taken:
|
Discussion: • Infrared-thermography papers paragraph (highlighted): Multi-task learning discussion citing Refs 31, 32, 38 with architectural details and DME application References: • Ref 31, 32, 38: Multimodal algorithm papers |
|
ADDITIONAL IMPROVEMENTS NOT REQUESTED BUT IMPLEMENTED |
|||
|
A.1 |
Reviewer 2 English language editing concern |
We performed thorough English language editing throughout the entire manuscript to improve:
This enhances overall manuscript quality and readability for international audiences. |
Throughout entire manuscript: |
Summary Statement:
We have systematically addressed all major and minor comments from both reviewers with comprehensive revisions. All requested analyses have been performed, appropriate statistical tools implemented (PROBAST-AI, DCA), critical methodological discussions added (clustering, high AUC values, IPD limitations), and all requested literature integrated with substantive discussion. Additionally, we performed extensive English language editing and data accuracy verification. The revised manuscript now provides robust, transparent, and comprehensive evidence for AI-based prediction of anti-VEGF treatment response in DME patients, with clear acknowledgment of limitations and detailed future research directions. We believe these revisions have substantially strengthened the manuscript and addressed all reviewer concerns comprehensively.
We sincerely thank both reviewers for their constructive feedback, which has significantly improved the quality and rigor of our manuscript.

Reviewer 2 Report
Comments and Suggestions for Authors
- The article systematically evaluates the diagnostic accuracy of AI in predicting anti VEGF treatment response in DME patients, but can further expand the search scope, supplement more currently published prospective cohorts or RCTs, and use individual case data (IPD) meta-analysis to improve the level of evidence and result stability.
- The AI model architecture, input modality, and hyperparameters used in the article have significant differences and moderate heterogeneity (I ² ≈ 45%). Further consideration can be given to establishing a multi center, publicly annotated DME-OCT benchmark dataset and conducting blind testing on all candidate models on this dataset to eliminate performance bias caused by device and population differences.
- The article can introduce decision curve analysis (DCA) and budget impact modeling (BIM) in subsequent research to compare the quality adjusted life years (QALY) and incremental cost-effectiveness ratio (ICER) of "AI assisted decision-making" and "conventional treatment" within 5 years, providing economic evidence for medical insurance payment and guideline revision.
4. Multimodal algorithms have great inspiration and reference value for this task, such as Lightweight bilateral network of Mura detection on micro-OLED displays, Multi-task learning for hand heat trace time estimation and identity recognition, Deep soft threshold feature separation network for infrared handprint identity recognition and time estimation. If the existing research in this article does not involve multimodal algorithms, the above-mentioned papers should also be mentioned. The article can further explore the hybrid 3D-CNN or Vision Transformer architecture of OCT and OCT-A fusion structures, verify whether vascular features such as blood flow density and non perfusion area volume can further improve the prediction of AUC, and explain their biological rationality.
Author Response

(The authors gave the same response as above.)

Round 2
Reviewer 1 Report
Comments and Suggestions for Authors
Reviewer 2 Report
Comments and Suggestions for Authors
The revised version has a very good improvement in algorithm and logic. I warmly recommend publication in present form.